# SHIFTED RANDOMIZED SINGULAR VALUE DECOMPOSITION

## ABSTRACT

We extend the randomized singular value decomposition (SVD) algorithm (Halko et al., 2011) to estimate the SVD of a shifted data matrix without explicitly constructing the matrix in the memory. With no loss in the accuracy of the original algorithm, the extended algorithm provides for a more efficient way of matrix factorization. The algorithm facilitates the low-rank approximation and principal component analysis (PCA) of off-center data matrices. When applied to different types of data matrices, our experimental results confirm the advantages of the extensions made to the original algorithm.

## 1 INTRODUCTION

The singular value decomposition (SVD) is one of the most used matrix decompositions in many areas of science. Among the typical applications of SVD are the low-rank matrix approximation and principal component analysis (PCA) of data matrices (Jolliffe, 2002). Using SVD to accurately estimate a low-rank factorization or the principal components of a data matrix, a mean-centering step should be carried out before performing SVD on the matrix. Despite its simplicity, the mean-centering can be very costly if the data matrix is large and sparse. This cost is because the mean subtraction of a sparse matrix turns it to a dense matrix which requires a considerable amount of memory and CPU time to be analyzed. This motivates us to extend the randomized SVD algorithm introduced by (Halko et al., 2011) to estimate the singular value decomposition of a mean-centered matrix without explicitly forming the matrix in the memory.

More generally, we introduce a shifted randomized SVD algorithm that provides for the SVD estimation of a data matrix shifted by any vector in the eigenspace of its column vectors. The proposed algorithm facilitates the low-rank matrix approximation and the principal component analysis of a data matrix through merging the mean-centering and the SVD steps. The mean-centering is crucial to obtain the minimum PCA reconstruction error through a deterministic SVD. We experimentally show that it plays an essential role in case of using the randomized SVD algorithm too. Our experiments with different types of data matrices show that the extended algorithm performs better than the original algorithm when both are applied to a center-off data matrix.

In the followings, we briefly introduce the principal component analysis and its connection with the singular value decomposition. Then, we introduce the shifted randomized SVD algorithm and provide an analysis of its performance. Finally, we report our experimental results obtained from the principal component analysis of different types of data matrices using the extended and the original randomized SVD algorithms.

## 2 PRINCIPAL COMPONENT ANALYSIS

Principal component analysis (PCA) is a method to study the variance of a random vector. PCA projects a random vector to the eigenspace of its covariance matrix. Let $\mathbf{x}$ be an $m$-dimensional random vector with the mean vector $\mathbf{0}$. PCA projects $\mathbf{x}$ to a latent random vector $\mathbf{y}$ as below:

$$\mathbf{y} = A^T \mathbf{x} \tag{1}$$

where the square matrix $A$ is composed of the eigenvectors of the covariance matrix $\Sigma_{\mathbf{x}}$. The elements of $\mathbf{y}$ are called the *principal components* of $\mathbf{x}$. In many use cases of PCA, $A$ contains

a subset of the eigenvectors of $\Sigma_{\mathbf{x}}$. The minimum PCA reconstruction error is obtained from the eigenvectors corresponding to the top eigenvalues of the covariance matrix (Jolliffe, 2002).

The matrix of eigenvectors $A$ in Equation 1 can be efficiently estimated from the singular value decomposition (SVD) of a sample matrix $X$. To this end, the sample matrix should be first centered around its mean vector:

$$\bar{X} = X - \mu_{\mathbf{x}}\mathbf{1}^T \tag{2}$$

The fact that the matrix of left singular vectors of $\bar{X}$ is equal to the eigenvectors of the covariance matrix of $X$, the PCA projection of $X$ is:

$$Y = U^T\bar{X} = SV^T \tag{3}$$

where $\bar{X} = USV^T$.

The mean-centering step in Equation 2 can be very costly if $X$ is a large sparse matrix and $\mu_{\mathbf{x}} \neq \mathbf{0}$. In this case, $\bar{X}$ is a dense matrix that requires a vast amount of memory and cannot be processed in a reasonable time. In the next section, we introduce a randomized SVD algorithm to estimate the SVD of $\bar{X}$ without explicitly performing the mean subtraction step.

## 3 SHIFTED SINGULAR VALUE DECOMPOSITION

Let $X$ be an $m \times n$ $(m \leq n)$ matrix and $\mu$ be an $m$ dimensional vector in the space of the column vectors of $X$. Algorithm 1, extends the randomized matrix factorization method introduced by Halko et al. (2011) to return a rank-$k$ approximation of the singular value decomposition of the matrix $\bar{X} = X - \mu\mathbf{1}^T$ without explicitly forming $\bar{X}$. The differences between the extended algorithm and the original one are in lines 6, 9, 10, and 12. In the followings, we explain the entire algorithm with a more in-depth focus on the modified parts.

The SHIFTED-RANDOMIZED-SVD algorithm works in three major steps:

1. Estimate a basis matrix for $\bar{X}$
2. Project $\bar{X}$ to the space of the basis matrix
3. Estimate the SVD factors of $\bar{X}$ from its projection

In the first step (lines 2 to 11), a rank $K$ basis matrix $Q_1$ $(k < K \ll m)$ that spans the column vectors of $\bar{X}$ is computed. In Line 2, a random matrix is drawn from the standard Gaussian distribution. This matrix is then used in Line 3 to form the sample matrix $X_1$ whose columns are independent random points in the range of $X$. This sample matrix is used to estimate a basis matrix for $\bar{X}$ in two steps. In Line 4, a basis matrix $Q_1$ is computed through QR-factorization of $X_1$. Since $X_1$ is sampled from $X$, the basis matrix is considered as an approximation of the basis of $X$. Then in Line 6, the basis of $\bar{X}$ is estimated from the $Q_1$ by the QR-update algorithm proposed by Golub & Van Loan (1996, p. 607). For a given QR factorization such as $Q_1R_1 = X_1$ and two vectors $u$, and $v$, the QR-update algorithm computes the QR-factorization in Equation 4 by updating the already available factors $Q_1$ and $R_1$.

$$QR = X_1 + uv^T \tag{4}$$

The computational complexity of the QR-update of the $m \times K$ matrix $X_1$ is $O(m^2)$.[1] Replacing $u$ with $-\mu$ and $v$ with $\mathbf{1}$, the QR-update in Line 6 returns the basis matrix $Q$ that span the range of the matrix

$$\bar{X} = X - \mu\mathbf{1}^T \tag{5}$$

In other words, $\bar{X}$ can be approximated from $Q$:

$$\bar{X} \approx QQ^T\bar{X} \tag{6}$$

Note that the basis matrix of $\bar{X}$ is computed without explicitly constructing the matrix $\bar{X}$ itself. The `if` statement in Line 4 controls the useless performance of the QR-update step with the null vector.

The `for` loop starting at Line 8, estimates a basis matrix for $B = (\bar{X}\bar{X}^T)^q\bar{X}$ using the basis of $\bar{X}$, $Q$. The matrix $B$ with a positive integer power has the same singular vectors as $\bar{X}$, but with a sharper

---

[1]The computational complexity of the QR-update of an $m \times n$ matrix in $O(N^2)$ where $N = \max(m, n)$.

decay in its singular values since $s_j(B) = s_j(\bar{X})^{2q+1}$, where $s_j(.)$ returns the $j$th singular vectors of its input matrix. The sharp decay in singular values improves the accuracy of the randomized SVD when the singular values of $\bar{X}$ decay slowly. This effect is because the reconstruction error of the randomized SVD is directly proportional to the first top unused singular vector of $\bar{X}$ (see Equation 12).

The basis of $(\bar{X}\bar{X}^T)^q \bar{X}$ is computed via alternative applications of matrix product on $\bar{X}^T$ and $\bar{X}$. For $q = 1$, in Line 9, a basis matrix of $\bar{X}^T \bar{X}$ is estimated through QR-factorization of $\bar{X}^T Q$. To avoid forming $\bar{X}$ explicitly, instead of direct multiplication $\bar{X}^T Q$, we use the distributive property of multiplication over addition:

$$\bar{X}^T Q = (X - \mu \mathbf{1}^T)^T Q = X^T Q - \mathbf{1}(\mu^T Q) \tag{7}$$

where $\mathbf{1}$ is a vector of ones. The product $\mathbf{1}(\mu^T Q)$ can be efficiently computed in $O(nK)$ memory space if a higher priority is given to the parentheses. In Line 10, a basis matrix of $\bar{X}\bar{X}^T \bar{X}$ is estimated through QR-factorization of $\bar{X} Q'$ where $Q'$ is a basis matrix of $\bar{X}^T Q$. Similar to Equation 8, the product $\bar{X} Q'$ is computed as:

$$\bar{X} Q' = (X - \mu \mathbf{1}^T) Q' = X Q' - \mu(\mathbf{1}^T Q') \tag{8}$$

with the same amount of memory space, $O(nK)$. The matrix multiplication loop iterates $q$ times. At this stage, we have the basis matrix $Q$ that approximates a basis for $\bar{X}$.

In the second major step of the algorithm, the matrix $\bar{X}$ is projected to the space spanned by $Q$:

$$Y = Q^T \bar{X} \tag{9}$$

This step in done in Line 12 using the same trick as in Equation 8:

$$Y = Q^T(X - \mu \mathbf{1}^T) = Q^T X - (Q^T \mu)\mathbf{1}^T \tag{10}$$

Finally, in the third step, the SVD factors of $\bar{X}$ are estimated from the $K \times n$ matrix $Y$ in two steps. First, a rank-$k$ approximation of $Y$ is computed using a standard method of singular value decomposition, i.e., $Y = U_1 \Sigma V^T$ (Line 13). Then, the left singular vectors are updated by $U = QU_1$ resulting in $U\Sigma V^T = QY$ (Line 14). Replacing $Y$ with $Q^T \bar{X}$ and using Equation 6 ($\bar{X} \approx QQ^T \bar{X}$), we have the rank-$k$ approximation of $\bar{X}$:

$$U\Sigma V^T \approx \bar{X} \tag{11}$$

---

**Algorithm 1** The rank-$k$ singular value decomposition of the $m \times n$ matrix $X - \mu \mathbf{1}^T = U\Sigma V^T$ with $(m \le n)$ using the sampling parameter $K$ $(k < K \ll m)$ and $q \in \{0, 1, 2, \dots\}$.

---
1: **procedure** SHIFTED-RANDOMIZED-SVD($X, \mu, k, K, q$)
2:      Draw an $n \times K$ standard Gaussian matrix $\Omega$
3:      Form the sample matrix $X_1 \leftarrow X\Omega$
4:      Compute the QR factorization $X_1 = Q_1 R_1$
5:      **if** $\mu \ne \mathbf{0}$ **then**
6:          Compute $QR = Q_1 R_1 - \mu \mathbf{1}^T$ using the QR-update algorithm
7:      **end if**
8:      **for** $i = 1, 2, \dots, q$ **do**
9:          Compute the QR-factorization $Q'R' = X^T Q - \mathbf{1}(\mu^T Q)$
10:         Compute the QR-factorization $QR = XQ' - \mu(\mathbf{1}^T Q')$
11:      **end for**
12:      Form $Y \leftarrow Q^T X - (Q^T \mu)\mathbf{1}^T$
13:      Compute the singular value decomposition of $Y = U_1 \Sigma V^T$
14:      $U \leftarrow QU_1$
15:      **return** $(U, \Sigma, V)$
16: **end procedure**

---

The shifting vector $\mu$ in Algorithm 1 can be any vector in the space of the column vectors of $X$. If it is set to the null vector $\mathbf{0}$, then the algorithm reduces to the original randomized SVD algorithm of Halko et al. (2011). If it is set to the mean vector of $X$, then the algorithm estimates the singular vectors of the mean-centered matrix $\bar{X}$. In this case, the algorithm facilitates the principal component analysis of a data matrix $X$ through merging the centering step in Equation 2 and the SVD step in Equation 3.

## 4   PERFORMANCE ANALYSIS

The SHIFTED-RANDOMIZED-SVD algorithm explained in the previous section approximates the SVD of a shifted data matrix $\bar{X} = X - \mu \mathbf{1}^T$ without explicitly constructing the matrix in the memory. In this section, we study the performance of the algorithm based on the accuracy and the efficiency of the original randomized SVD algorithm of Halko et al. (2011).

To estimate the singular value decomposition of a sifted matrix $\bar{X}$ using the original randomized SVD algorithm, $\bar{X}$ should be explicitly formed and passed to the algorithm. Since SHIFTED-RANDOMIZED-SVD adds no extra randomness to the original algorithm, we have the same reconstruction error bound as if the original algorithm factorized the shifted matrix $\bar{X}$:

$$\mathbf{E}[\|\bar{X} - USV^T\|] \leq \left[1 + 4\sqrt{\frac{2m}{k-1}}\right]^{\frac{1}{2q+1}} \sigma_{k+1} \tag{12}$$

where $\sigma_{k+1}$ is the $(k+1)$th singular value of the $m \times n$ matrix $\bar{X}$ with $m \leq n$, $2 \leq k \leq \frac{m}{2}$ is the decomposition rank, and $q \in \mathbb{Z}^+$ is a power value as explained in Algorithm 1.

In the followings, we study the computational complexity of the SVD factorization of $\bar{X}$ using the original randomized SVD algorithm and its extended version in Algorithm 1. For an $m \times n$ matrix $\bar{X}$, the computational complexity of the original randomized SVD algorithm is:

$$O(\alpha k + (m + n)k^2) \tag{13}$$

where $\alpha$ is the cost of the matrix-vector multiplication with the input matrix $\bar{X}$. If $\bar{X}$ is a dense matrix then $\alpha = mn$, and if $\bar{X}$ is a sparse matrix then $\alpha = T$, a small constant value.

Algorithm 1 adds a QR-update step (Line 6) and three matrix-matrix multiplications (lines 9, 10, and 12) to the original algorithm. The matrix multiplications do not affect the computational complexity of the original algorithm since their computational complexity is equal to the complexity of computing $Q^T X$ in the original algorithm. The QR-update step, running in $O(m^2)$, however, can affect the computational complexity of the algorithm.

Assuming that $\mu \neq \mathbf{0}$, if both $X$ and $\bar{X}$ are dense matrices then both algorithms have the same computational complexity as:

$$O(mnk + (m + n)k^2) \tag{14}$$

This equality is because the computational complexity of the QR-update step, $O(m^2)$, is dominated by the complexity of the original algorithm (i.e., $m^2 \leq mn$ for every $m \leq n$ where $m, n \in \mathbb{N}$). In addition, the construction of $\bar{X}$ to be use by the original algorithm takes $O(mn)$ time which is greater than or equal to the complexity of the QR-update step. Hence, the added operations do not affect the computational complexity of the original algorithm. Halko et al. (2011) show that for a dense input matrix, the randomized SVD algorithm can be performed in $O(mn \log k + (m+n)k^2)$ if instead of the random normal matrix $\Omega$ in Line 2, a structured random matrix such as the sub-sampled random Fourier transform is used. This improvement can be considered for the SHIFTED-RANDOMIZED-SVD algorithm too.

If the input matrix $X$ is sparse, then $\bar{X}$ is a dense matrix for every $\mu \neq \mathbf{0}$. In this case, the computational complexity of the SHIFTED-RANDOMIZED-SVD algorithm is:

$$O(Tk + m^2 + (m + n)k^2) \tag{15}$$

where the constant $T$ is the cost of multiplying a sparse matrix to a vector, and the parameter $m^2$ is related to the complexity of the QR-update step. On the other hand, since $\bar{X}$ is a dense matrix, the complexity of the original algorithm is $O(mnk + (m + n)k^2)$ which is higher than the complexity of the extended algorithm.

In a special case where $X$ is a dense matrix and $\bar{X}$ is a sparse matrix, the original algorithm can factorize $\bar{X}$ in $O(Tk+(m+n)k^2)$, but SHIFTED-RANDOMIZED-SVD needs $O(mnk+(m+n)k^2)$ time. In this case, if Algorithm 1 is applied to $\bar{X}$ with $\mu = \mathbf{0}$, the factorization can be performed in the same processing time as the original algorithm. As a summary, we showed that the SHIFTED-RANDOMIZED-SVD algorithm as illustrated in Algorithm 1 is as efficient as the randomized SVD algorithm proposed by Halko et al. (2011) if the input matrix is dense, and more efficient than it if the input matrix is sparse.

## 5 EXPERIMENTS

We experimentally study the difference between performing PCA with the randomized SVD algorithm (RSVD) proposed by Halko et al. (2011) and its extended version in Algorithm 1 (S-RSVD). The fact that a minimum PCA reconstruction error is obtained from the deterministic SVD of a mean-centered data matrix, the performance of S-RSVD on an off-center data matrix with its mean vector as the shifting vector is expected to be more accurate than the performance of RSVD on the same data matrix. Since the two algorithms are randomized in nature, it is important to test if this expectation is valid for different types of data matrices.

We experimentally compare the two algorithms based on the mean of the squared $L_2$ norm of PCA reconstruction error (MSE). The same parameters $K = 2k$ and $q = 0$ are used for both S-RSVD and RSVD unless it is clearly mentioned. The shifting vector $\mu$ for S-RSVD is set to the mean vector of data matrices. The experiments are carried out on different types of data matrices including randomly generated data, a set of images, and word co-occurrence matrices. The characteristics of the data matrices are illustrated in the corresponding sections.

### 5.1 RANDOM DATA

In this section, we examine how the two SVD algorithms are affected by the parameters such as the number of principal components, the size and the distribution of a data matrix, and the power iteration scheme. Our experiments are based on two comparison metrics, 1) an MSE value obtained from a fixed number of principal components, and 2) the sum of MSE values obtained from different number of principal components ranging from 1 to 100.

Figure 1a represents the effect of the number of principal components on the MSE values obtained from a $100 \times 1000$ matrix sampled from a 100-dimensional random vector uniformly distribution in the range $\mathbf{0}, \mathbf{1}$. The results show that mean-centering leads to substantial reduction to the reconstruction error when the number of principal components is small. This observation is in line with the fact that the contribution of the mean-centering is mostly to the accuracy of top principal components.

The effect of the sample size on the two factorization algorithms is represented in Figure 1b in which the X-axis is the sample size and the Y-axis is the sum of the MSE values obtained from different number of principal components ranging from 1 to 100. The data matrices are generated by a 100-dimensional uniform random vector in the range $\mathbf{0}, \mathbf{1}$. The results show that S-RSVD is more accurate and stable than RSVD. Despite the fact that both algorithms are randomized, the stability of S-RSVD is less sensitive to the sample size.

Figure 1c compares the performance of the two algorithms with respect to the data distribution. The Y-axis is the sum of MSE values.Regardless of the data distribution, S-RSVD is more accurate than RSVD. This observation is in line with the fact that PCA does not make any assumption about the data distribution.

To examine whether both algorithms are equally accurate for the factorization of a mean-centered data matrix $\bar{X}$, a comparison between the two algorithms is provided in Figure 1d. The Y-axis is the sum of the MSE values.In the experiments with RSVD, the matrix $\bar{X}$ is explicitly constructed and factorized, but in the S-RSVD experiments the singular factors of $\bar{X}$ are implicitly estimated from $X$. The results show that S-RSVD is as accurate as RSVD applied to an already centered data matrix $\bar{X}$. This observation supports Equation 11.

An important parameter of the randomized SVD algorithm is the power value $q$ that has a positive effect on the accuracy of the algorithm, . Figure 1e shows the MSE values obtained from each of the factorization algorithms with different values of $q$. The data matrix in this experiment is sampled from a 100-dimensional uniform distribution. The Y-axis is the sum of MSE values and the X-axis represents $q$. The importance of mean-centering is clear when the value of $q$ is small. The accuracy of RSVD is significantly improved as the values of $q$ increases, while the accuracy of S-RSVD is only slightly improved. This observation on a set of uniformly distributed vectors suggest that RSVD with a positive value of $q$ (1 or 2 as suggested by Halko et al. (2011)) can be as accurate as S-RSVD.

To test whether RSVD with a large value of $q$ can be as accurate as S-RSVD, we run the same experiment as above but on data with different distributions. Figure 1f shows the difference between

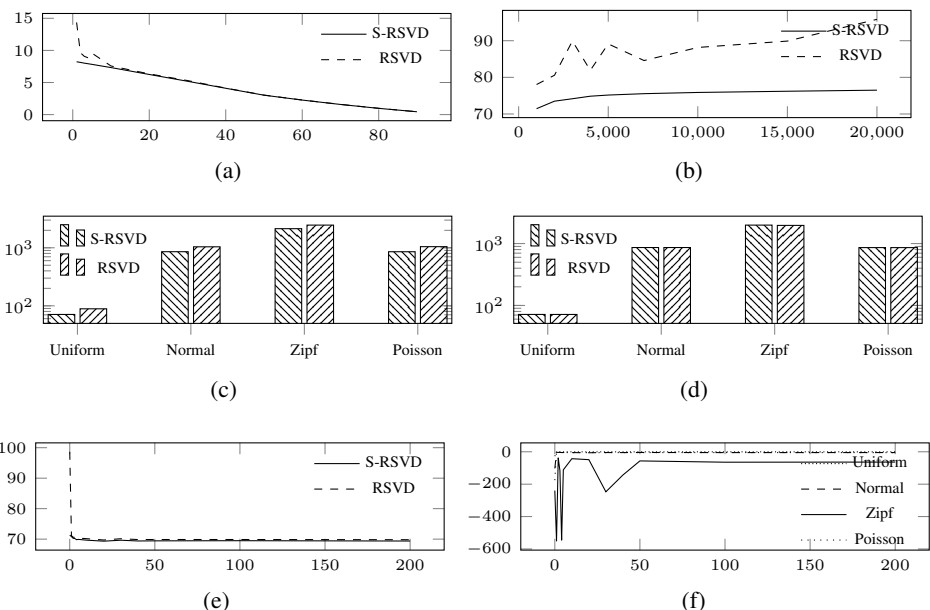

Figure 1: A comparison between S-RSVD and RSVD based on (a) number of principal components, (b) sample size, (c) data distribution, (d) explicit versus implicit mean-centering, (e) the power value $q$, and (f) the difference of their accuracies with respect to the power value $q$ and the data distribution.

the sum of MSE values obtained from each of the algorithms (i.e., Y-axis is MSE-SUM(S-RSVD) − MSE-SUM(S-RSVD)) with respect to the parameter $q$. Being all the results negative means that S-RSVD is more accurate than RSVD. Except for the data with Zipfian distribution, the difference between the accuracy of the two algorithms approaches to zero as the value of $q$ increases. The Zipfian graph fluctuates widely for small values of $q$, but it becomes flat as $q$ becomes larger. In the best case, the difference between the two algorithms on Zipfian data is $-64$ at $q = 200$. This indicates that the power iteration scheme cannot fully recover the reconstruction loss of an off-center data matrix, but depending on the data distribution, it can be helpful. The power iteration in Algorithm 1 is a computationally heavy step which can negatively affect the efficiency of the algorithm when the value of $q$ is large.

## 5.2 IMAGE DATA

In this section, we experiment with handwritten digits and facial image matrices. The handwritten digits are a copy of the test set of the UCI ML hand-written digits datasets consisting of 1979 images of size $8 \times 8$.[2] We vectorize individual image matrices and stack them into a single $16 \times 1979$ data matrix. The facial images consisting of 13233 images each of size $250 \times 250$ are downloaded from Labeled Faces in the Wild (LFW).[3] The facial image matrix after vectorizing and stacking all of the images matrices is a $62500 \times 13233$ matrix.

The left side of Table 1 summarizes the results obtained from 10-dimensional PCA of the image matrices. The MSE values represented in the first two rows of the table show that S-RSVD is more accurate than RSVD. To ensure that the results are not due to chance, we run the experiment 30 times and perform two t-tests with the following null hypotheses:

- $H_0^1$:*there is no difference between the MSE of S-RSVD and RSVD.*
- $H_0^2$:*there is no difference between the individual column reconstruction errors of S-RSVD and RSVD.*

---

[2]https://archive.ics.uci.edu/ml/datasets/Optical+Recognition+of+Handwritten+Digits
[3]http://vis-www.cs.umass.edu/lfw/lfw.tgz

| | image data | | word data | | | |
| --- | --- | --- | --- | --- | --- | --- |
| | digits | faces | $n = 1\text{e}3$ | $n = 1\text{e}4$ | $n = 1\text{e}5$ | $n = 3\text{e}5$ |
| MSE of S-RSVD | **415.7** | **15.3**e7 | **195**e$-$**5** | **235**e$-$**5** | **763**e$-$**5** | **994**e$-$**5** |
| MSE of RSVD | 430.6 | 16.1e7 | 200e$-$5 | 236e$-$5 | 765e$-$5 | 998e$-$5 |
| $p_1$-value | 0.00 | 0.00 | 0.00 | 0.00 | 0.00 | 0.00 |
| $p_2$-value | 0.00 | 0.00 | 0.00 | 0.00 | 0.00 | 0.00 |
| WR of S-RSVD | **66**% | **82**% | **71**% | **73**% | **77**% | **70**% |
| WR of RSVD | 34% | 18% | 29% | 27% | 23% | 30% |

Table 1: The reconstruction error statistics of image and word data.

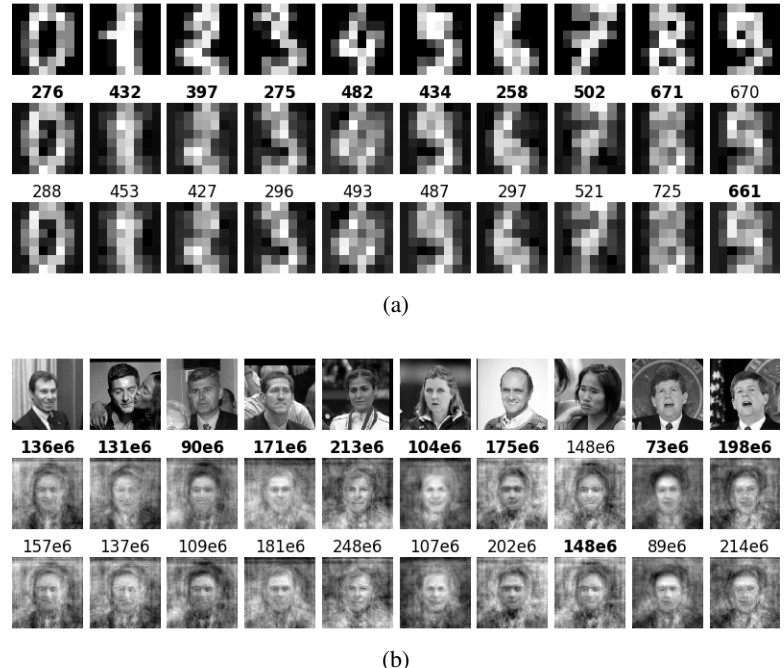

Figure 2: The effect of mean-centering on (a) handwritten digits and (b) facial data. In each sub-figure, the top row is the original image, the rows in the middle and bottom are the S-RSVD and RSVD reconstructed images, respectively. The reconstruction errors are shown on top of each image.

The former hypothesis is validated on the 30 MSE pairs obtained from the SVD methods, but the later hypothesis is validate on the pairs of the reconstruction error of individual images. The $p$-values represented in Table 1 reject both hypotheses and confirm that the better results obtained from S-RSVD are not by chance. The rejection of $H_0^2$ indicates that S-RSVD results in not only lower MSE for the entire image matrices, but also for individual images.

To provide a better picture of how well the SVD algorithms perform on individual images, we plot the first 10 images of each data matrix and estimate the win-rates of the algorithms. Figure 2 shows the examples of the original handwritten and facial images (the top rows), and their reconstructions using S-RSVD (the middle rows) and RSVD (the bottom rows) with the reconstruction error values on top of each image. For most images, S-RSVD is more accurate than RSVD. To generalize this observation, we estimate the win-rate (WR) of the algorithms (i.e., the number of images for which one algorithm is more accurate than the other algorithm out of the total number of images). The results shown in Table 1 indicate that 66% of the handwritten images and 82% of the facial images are reconstructed more accurately by S-RSVD than RSVD.

### 5.3 Word Data

In this section, we experiment with word probability co-occurrence matrices whose elements are the probability of seeing a target word in the context of another word. Our experiments are based on the word co-occurrences probabilities estimated from the English Wikipedia corpus used in the CoNLL-Shared task 2017.[4] For each target word $w_i$, we estimate the probability of seeing the word conditioned on the occurrence of another word $w_j$, called the context word (i.e., $p(w_i|w_j) \approx \frac{n(w_j, w_i)}{n(w_j)}$). The $i$th column of a probability co-occurrence matrix associated with the word $w_i$ is a distributional representation of the word.

Due to the Zipfian distribution of words and relatively large number of words, a word probability co-occurrence matrix is a large and sparse matrix with a high degree of sparsity. A mean subtraction turns the matrix to a dense matrix that needs huge amount of memory and processing time to be analyzed. The SHIFTED-RANDOMIZED-SVD algorithm can substantially improve the performance of analyzing a mean-centered co-occurrence matrix.

We estimate 100-dimensional PCA representations of different $m \times n$ word probability co-occurrence matrices formed with $m = 1000$ most frequent context words and $n$ most frequent target words with different values of $n$. Each experiment is run 30 times with different random seeds. The right side of Table 1 represents the statistics of the reconstruction errors obtained from each of the factorization algorithms. The first two rows of the table confirm that S-RSVD is more accurate than RSVD. To see whether the difference between MSEs is statistically significant, a t-test with the null hypothesis $H_0^1$:*there is no difference between the MSE of S-RSVD and RSVD* is performed. Using the 30 pairs of MSE values obtained from each of the factorization algorithms, the test rejects the null hypothesis $H_0^1$ with a high confidence level (see $p_1$-value in Table 1).

We study the effect of the mean-centering on the reconstruction of the distributional representation of individual words (i.e., each column of the co-occurrence matrix). A t-test is performed to validate the null hypothesis $H_0^2$:*there is no difference between the individual column reconstruction errors of S-RSVD and RSVD*. The acceptance probabilities of $H_0^2$ shown as $p_2$-values in Table 1 confirm that the differences between the reconstruction errors of individual words is indeed significant. The win-rates (WR) of each of the algorithms shows that the mean-centering is beneficial to the reconstruction of the majority of words.

### Conclusion

We extend the randomized singular value decomposition algorithm of Halko et al. (2011) to factorize a shifted data matrix (i.e., a data matrix whose columns vectors are shifted by a vector in their eigenspace) without explicitly constructing the matrix in the memory. With no harm to the performance of the original algorithm on dense matrices, the extended algorithm leads to substantial improvement to the accuracy and efficiency of the algorithm when used for low-rank approximation and principal component analysis of sparse data matrices. The algorithm is tested on different types of data matrices including randomly generated data, image data, and word data, with their mean vector as the shifting vector. The experimental results show that the extended algorithm results in lower mean squared reconstructions error in all experiments through successfully incorporating the mean-centering step to SVD.

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
