# OpenReview forum: "Shifted Randomized Singular Value Decomposition"
_ICLR.cc/2020/Conference — Reject_

### Official Review · AnonReviewer2 · 2019-10-23
**Official Blind Review #2**

**Rating:** 1

**Review:**

This paper adapts the approach by Halko to get a  SVD using
a low rank concept to the case where the matrix is implicit shifted.
Honestly - there is nothing wrong with this paper except the level
of contribution. I consider this work to be widely irrelevant. You
can report this on arxiv if you like but I do not think it is important in general.
The results show some effect - but not a relevant one.
For ICLR this is much to less. And there is not much more to say.

-----------------------------------------------------------------------------------
-----------------------------------------------------------------------------------
-----------------------------------------------------------------------------------
-----------------------------------------------------------------------------------
-----------------------------------------------------------------------------------
-----------------------------------------------------------------------------------
-----------------------------------------------------------------------------------
-----------------------------------------------------------------------------------
-----------------------------------------------------------------------------------
-----------------------------------------------------------------------------------
-----------------------------------------------------------------------------------
-----------------------------------------------------------------------------------


**Experience Assessment:**

I have published in this field for several years.

**Review Assessment: Checking Correctness Of Derivations And Theory:**

I assessed the sensibility of the derivations and theory.

**Review Assessment: Checking Correctness Of Experiments:**

I assessed the sensibility of the experiments.

**Review Assessment: Thoroughness In Paper Reading:**

I read the paper thoroughly.

---

### Official Review · AnonReviewer3 · 2019-10-23
**Official Blind Review #3**

**Rating:** 1

**Review:**

This paper proposes a method for performing an SVD on a mean-subtracted matrix without having to actually subtract the mean from the matrix first.

While the problem of calculating an SVD is fundamental, the paper’s idea and general problem it covers is a clear mismatch for ICLR in my opinion. The algorithm is an extension of a previous one. Since the idea is very simple, one would expect a lot of theory, but the main theoretical result in Section 4 can be copy-pasted from the previous algorithm since they share the same result. Three sources are cited, indicating either a very narrow view of the field, or overconfidence in the fundamental significance of the contribution.

The experiments show the technique works as advertised, but the importance of the result is low.

**Experience Assessment:**

I have read many papers in this area.

**Review Assessment: Checking Correctness Of Derivations And Theory:**

I assessed the sensibility of the derivations and theory.

**Review Assessment: Checking Correctness Of Experiments:**

I assessed the sensibility of the experiments.

**Review Assessment: Thoroughness In Paper Reading:**

I read the paper at least twice and used my best judgement in assessing the paper.

---

### Official Review · AnonReviewer1 · 2019-10-24
**Official Blind Review #1**

**Rating:** 3

**Review:**

This paper extends the existing randomized SVD algorithm by Halko et al. to propose a shifted randomized SVD algorithm. The proposed algorithm performs randomized SVD on \bar{X}=X-\mu 1^T with a given constant vector \mu without explicitly constructing \bar{X} from X.

The proposed algorithm seems to be a straightforward modification of the randomized SVD algorithm, so that the contribution of this paper would be incremental. Also, the claimed benefits of the proposed method are better accuracy, less memory usage, and less computational time. On memory usage and computational time, however, I could not find any experimental comparison in Section 5. At least in Section 5.3 where the data matrix is indeed sparse, there should be some comparison in these respects. Because of these reasons, I would not be able to recommend acceptance of this paper.

In Algorithm 1, line 6, I think that one has to subtract not \mu 1^T but \mu 1^T \Omega.

I guess that equation (12) comes from Theorem 1.2 in Halko et al. (2011). An explicit citation should be needed here to put an appropriate credit to that paper. Also, the conditions under which equation (12) holds should be explicitly stated. In fact, Theorem 1.2 in Halko et al. is about the expected reconstruction error with rank-2k factorization obtained by randomized SVD, which is not consistent with the authors claim in page 3, lines 4-5: The first top unused singular value is not \sigma_{k+1} but \sigma_{2k+1}.

Page 1, line 25: that provides (for) the SVD estimation
Page 2, line 38: that span(s) the range
Page 4, line 6: a s(h)ifted matrix
Page 4, line 29: to be use(d) by the
Page 5, line 22: uniformly distribut(ion -> ed)
Page 5, lines 33, 38: Space is missing after the period.
Page 5, line 43: There is an extra comma.
Page 5, line 48: suggest(s) that
Page 6, line 4: approaches (to) zero
Page 6, line 14: 16 \times 1979 should read 64 \times 1979.
Page 7, line 2: is validate(d)


**Experience Assessment:**

I do not know much about this area.

**Review Assessment: Checking Correctness Of Derivations And Theory:**

I assessed the sensibility of the derivations and theory.

**Review Assessment: Checking Correctness Of Experiments:**

I assessed the sensibility of the experiments.

**Review Assessment: Thoroughness In Paper Reading:**

I read the paper at least twice and used my best judgement in assessing the paper.

---

### Decision · Program_Chairs · 2019-12-19

**Decision:**

Reject

**Comment:**

The proposed algorithm is found to be a straightforward extension of the previous work, which is not sufficient to warrant publication in ICLR2020.